# Perception of the Role of Food and Dietary Modifications in Patients with Inflammatory Bowel Disease: Impact on Lifestyle

**DOI:** 10.3390/nu13030759

**Published:** 2021-02-26

**Authors:** Laura Guida, Francesca Maria Di Giorgio, Anita Busacca, Lucio Carrozza, Stefania Ciminnisi, Piero Luigi Almasio, Vito Di Marco, Maria Cappello

**Affiliations:** Gastroenterology and Hepatology Section, PROMISE, University of Palermo, Piazza delle Cliniche, 2-90127 Palermo, Italy; laura.guida91@gmail.com (L.G.); francescam.digiorgio@libero.it (F.M.D.G.); anitabusacca05@gmail.com (A.B.); lucio.carrozza@gmail.com (L.C.); stefaniaciminnisi@yahoo.it (S.C.); piero.almasio@unipa.it (P.L.A.); vito.dimarco@unipa.it (V.D.M.)

**Keywords:** inflammatory bowel disease, dietary changes, food perception

## Abstract

Background: Diet has a relevant role in triggering symptoms in inflammatory bowel disease (IBD) from the patients’ perspective, but there is gap the between patients’ and doctors’ perceptions. Few studies have addressed this topic. The aim of this study was to evaluate food habits and nutrition knowledge in a homogeneous cohort of patients with IBD from southern Italy. Methods: 167 consecutive patients with IBD were recruited. The survey was based on the administration of a semi-structured questionnaire assessing demographics, disease features, dietary behavior, and food intolerance. Results: The majority of patients did not consider food a cause of their disease. However more than 80% changed their diet after the diagnosis and most report an improvement in symptoms. Spiced and seasoned foods, dairy products, vegetables, and fruit were often avoided. A dairy-free diet was adopted by 33.7%. Food choices were based on self-experience and not on medical counselling. Dietary modifications deeply impact on lifestyle. Conclusions: Most of the patients with IBD set diet and lifestyle on self-experience and give up many foods. This has an impact on psychosocial functioning and can lead to nutritional deficiencies. High quality studies are warranted to assess evidence-based dietary strategies and develop patient-targeted dietary recommendations.

## 1. Introduction

Chronic inflammatory bowel disease (IBD) is a chronic disease of the gastrointestinal tract that includes two main conditions: Crohn’s disease (CD), which can involve any part of the digestive tract, and ulcerative colitis (UC), which affects only the colon. Both are more prevalent in Western countries, although incidence and prevalence are increasing worldwide [1].

The etiopathogenesis of these diseases is unknown, but available evidence suggests an interaction of genetic predisposition, the immune system, the intestinal microbiota, and the function of the intestinal epithelial barrier and the environment. Among environmental risk factors, cigarette smoking has been demonstrated to be positively related to the onset of CD, while it is a protective factor in UC. The higher prevalence of IBD in Western countries and the increase in incidence in developing countries has led to the hypothesis of a role of diet in relation to Westernization [1,2,3].

Previous studies [4,5,6,7,8,9] have shown that most patients believe that food is as important as IBD medications in symptoms management. Following diagnosis, many patients change their diet, aiming at reducing symptoms and prolonging the period of remission of the disease, and they often adopt restrictive diets that can lead to malnutrition and a significant reduction in quality of life. Among the foods most frequently avoided are spicy foods, milk and dairy products, heavily seasoned foods, carbonated drinks, energy drinks, fried foods, and alcoholic beverages. Furthermore, many patients try to eat meals regularly, tend to reduce portions, and take food supplements. Ethnicity has been found to be significantly related to strong beliefs on the role of diet in the studies from the Manchester IBD group. Moreover, half of the patients had never received dietary advice and two thirds wanted to receive more food-related information [6]. In an Australian study [7], 61% of members of a large patient support group believed that their IBD specialist disregarded the importance of diet. In a more recent survey from Manchester conducted in patients with inactive UC, 90% of subjects declared they relied on their own experience and only 10% received advice from a healthcare professional [5].

No significant differences have been observed between patients with UC and patients with CD in most studies, with only one study [5] reporting a lower perception of the role of diet in UC. Compared to men, women tend to attribute the same importance to nutrition and therapy, and also tend to eliminate food and use food supplements more often than men do [4,8]. In some cases, specific diets such as gluten or lactose-free diets are adopted [10,11,12].

However, studies are conflicting, have variable sample sizes and methodology, most come from northern Europe or North America, and very few have been conducted in southern Europe and in populations on a Mediterranean diet [4,5,6,7,8,9,10,11,12].

The aim of our study was to investigate food habits and nutrition knowledge as well as the impact of dietary changes on lifestyle in a homogeneous cohort of patients with IBD from southern Italy.

## 2. Materials and Methods

Patients with IBD coming to the clinic dedicated to IBD of the Gastroenterology and Hepatology Unit of the University Hospital of Palermo for follow-up visits or for the infusion of biological drugs from September 2019 to June 2020 were recruited. They were asked for informed consent to participate in the study. The study was conducted according to the guidelines of the Declaration of Helsinki and approved by the local Ethics Committee: the ethical approval code is 11/2020. The study involved the administration of a semi-structured questionnaire consisting of 48 questions, divided into four sections: A, B, C, D (see Appendix A). The questionnaire was developed by the Authors and reviewed by a hospital patients’ focus group with IBD and the board of the local School of Dietetics. The interview was conducted by the clinicians of the research team and lasted 20–30 min.

Section A “Personal data and basic information” contained 15 questions relating to personal data, weight, height, ethnicity, marital status, educational qualification, and employment status.

Section B “Characteristics of the disease” contained six questions relating to the diagnosis, the duration of the disease in years, the disease activity perceived by the patient at the time of the interview, current therapy, and previous surgery.

Section C “Eating habits” consisted of 17 questions: patients were asked if they consider nutrition as one of the causes of the disease and if they have changed their diet (Have you changed your diet since diagnosis? What happened to symptoms following dietary changes? Have you found any foods or drinks that trigger the symptoms? Have you avoided some foods since diagnosis? Do you always avoid them or only during the activity or remission phase?) and their lifestyle following diagnosis (Have you given up on activities in your daily life due to illness? Has the disease affected your social life (i.e., going out to eat with family or friends)? Has your lifestyle changed due to illness?).

Section D “Food intolerances” was made up of 10 questions; patients were asked if they are on a gluten-free or lactose-free diet and the reasons for this choice (Do you follow a gluten-free diet? Have you ever tested for gluten intolerance? If you have never performed any tests, why are you on a gluten-free diet? What symptoms do you experience after eating gluten-containing foods? Are you on a lactose-free diet? Have you ever been tested for lactose intolerance? If you have never been tested, why are you following a lactose-free diet? What do you mean for lactose-free diet? What symptoms do you experience after eating lactose-containing foods?).

For each patient, a form was also filled by the interviewers that reported the Partial Mayo Score for UC [13] and HBI [14] (Harvey Bradshaw Index) for CD for the assessment of disease activity at the time of the interview. Weight, height, and body mass index (BMI), were also recorded.

Data (perception of the role of foods, list of foods avoided, specific diets) were analyzed on the whole population.

Statistical analysis was performed with SPSS version 25.0 (SPSS Inc., Chicago, IL, USA). Continuous variables were expressed as mean ± SD, categorical variables as rate and percentage. Comparison between continuous variables was assessed using the Student’s *t* test or the Mann–Whitney’s test, while the Chi square test (χ^2^) was used for categorical variables. A *p*-value ≤ 0.05 was considered statistically significant.

## 3. Results

A total of 167 consecutive patients with IBD were enrolled: 84 patients were undergoing biological therapy (Infliximab, Vedolizumab, Ustekinumab) and were interviewed in the infusion room and 83 patients were interviewed in the examination rooms. 

Eighty-one patients (48.8%) had UC, 86 (51.5%) had CD (baseline characteristics in terms of demographic and clinical features of the whole population studied are reported in Table 1). Mean age was 48.6 ± 16 (range 18–77 years), 57.5% were males. Most patients (65.9%) were married, most were non-smokers (41.9% never smoked, 37.7% ex-smokers), and mean BMI was 24.4 ± 3.9. Diagnosis of IBD had been formulated in the past 10 years in 60%, while disease duration was longer than 20 years in 19.8%. In terms of educational status, 10% had only attended primary school, 39% had a secondary school degree, 36.5% a high school degree, and 14.4% a university qualification. Regarding occupational status, 6.6% were students, 46.2% had a fulltime job, 16.2% were unemployed, 16.8% retired, and 19.8% were homemakers. 

Most patients were in remission or had mild disease activity, as assessed by clinical scores (Partial Mayo Score and HBI), but patients perceived their disease as more severe than clinicians (Table 1). 

Regarding medical therapy, 83.8% of patients were on mesalamine, 14.4% on conventional immunomodulators, 20.4% on steroids, while 50.3% were on biologics; the most used was infliximab (52.4%) followed by vedolizumab (45.2%) and ustekinumab (2.4%).

There were no significant differences between the two groups of patients at baseline except in age, higher in patients on conventional therapy (though not significant; *p* = 0.02), and BMI, which was higher in patients on biologics (Table 2).

When asked if they considered diet as the cause of their disease, 59.28% of the study population did not believe that food was the main cause of IBD, while 40.7% did. When we analyzed the differences between those who answered “No” and those who answered “Yes”, there was a significant difference in dietary modifications and in the perceived clinical benefit obtained with such modifications in the second group (Table 3).

When patients were asked if they modified their diet after diagnosis of IBD, 82% answered they had changed their diet, and again this was significantly related to perception of a causative role of diet; use of specific diets was also significantly more common in the group who decided to change their diet. There were no differences in baseline characteristics and concomitant therapy (Table 4). At least 65% of patients reported a clinical benefit from dietary modifications.

We found 27.5% of patients adopted a specific diet (Table 5). The only variable significantly associated with adoption of specific diets (lactose-free or gluten-free or low-fiber) was female gender. A lactose-free diet was adopted by 33.6% of patients, but only 16.9% had done a specific test such as a breath test or assessment for allergy to milk protein to diagnose lactose intolerance or milk protein hypersensitivity. The reason for avoiding milk and dairy products was their suspected role in triggering symptoms (in 67.9% of patients), while a lactose-free diet was advised by the treating physicians only in 14.3% of patients. A lactose-free diet was more common in patients with CD (*p* = 0,08). 

A gluten-free diet was adopted only by 7 patients (4.2%), 3 of them had celiac disease, the others believed gluten was a trigger of symptoms. In none of these patients was a gluten-free diet advised by doctors.

Most patients, about 80%, and especially patients with CD (see Table 6) avoided certain foods considered as triggers, and this avoidance was usually practiced in both exacerbations of the disease and remission periods. This occurred upon medical advice in a minority of patients, especially for vegetable avoidance (13.2%). Foods more frequently avoided were spicy foods, seasoned foods, fried foods, milk and dairy products, carbonated drinks, spirits, vegetables, legumes, and whole grain bread. Processed meat was avoided in about 6.6% (only in 1.8% upon medical advice) and coffee in 12.6%.

The use of nutritional supplements was reported by 25% of patients.

Most patients were satisfied of their body weight (64%), 41% had a reduction of body weight from the time of diagnosis, while 33% reported an increased body weight since diagnosis.

The disease as well as dietary changes had an impact on lifestyle: 48% of patients reduced physical exercise, especially outdoors (for fear of symptoms, especially diarrhea), and from 39.8 to 53.6% reduced their opportunities of social life (eating out, meeting friends). Most patients reported they try to avoid going out for dinner or entertainment and, if they do, they want to be reassured of the availability of certain foods or access to toilets. There were no differences between patients with UC or CD, while there was a significant difference in reducing social activities for patients on biologics (*p* = 0.07).

## 4. Discussion

We carried out a survey on food habits and perception in a homogeneous cohort of patients with IBD treated in a tertiary referral center in Southern Italy. This is, to our knowledge, the only study addressing these topics in a Mediterranean cohort.

Our study confirmed that patients with IBD, regardless of their ethnic origin, believe diet has a relevant role in the onset and the clinical course of IBD: a significant proportion of our cohort believes diet or certain foods are the cause of their disease and have modified their diet according to these beliefs; others think diet is not the main cause but can significantly impact on the course of the disease, since some foods can trigger clinical symptoms. Our results are comparable with those of previous studies conducted in northern European or North American populations [4,5,6,7,8,9]. Every patient has collected his or her own list of prohibited foods and reports a clinical benefit from avoiding such foods. Spicy foods, seasoned and fried foods, carbonated drinks, and dairy products are on top of the list. In addition, legumes and vegetables, the cornerstones of the Mediterranean diet, are avoided and this occurs not only during flare-ups but also in remission, for fear of inducing relapses. Cohen, et al. [15] reported that patients with active disease had different dietary patterns than those with a quiescent disease. Moreover, the same Authors reported that UC patients were able to consume more raw vegetables. In contrast, there were no differences in our cohort according to disease type or activity. These discrepancies could be explained by several reasons: most of our patients were in remission or with mild activity, UC and CD were equally represented, while more patients with CD completed the survey in the study by Cohen et al., [15] An increased consumption of vegetables in patients with UC as compared to those with CD was reported also by Zallot et al. [4].

Previous studies [16,17] showed that the incidence of IBD is higher in patients who do not eat fruit and vegetables but eat a lot of fats and proteins. Recently, the International Organization for the Study of IBD (IOIBD) recommended increasing exposure to fruit and vegetables in Crohn’s disease [18]. Moreover, a specific food pyramid [19], maintaining complex carbohydrates every day together with fruits and vegetables has been designed for IBD by Italian researchers. The risk of inadequate folate intake and its consequences related to the avoidance of vegetables should be also considered. Other studies [20] have shown that the Mediterranean diet is associated with a better course of CD and, although guidelines do not recommend a specific diet for IBD, the Mediterranean diet should be preserved.

The elimination of fibers or a low-fiber intake in patients with IBD, even when in remission, was a common dietary practice in our cohort. This practice should not be encouraged for several reasons. Epidemiological studies [16] have evaluated the association between fiber intake and the development of IBD in European population. Fibers could prevent the development of IBD: they reduce intestinal transit time, are converted by colon bacteria into short-chain fatty acids, including butyrate, which is the main energy substrate for colonocytes, can reduce intestinal inflammation, affect the composition of the intestinal microbiota, and finally are involved in maintaining the functioning of the intestinal barrier. Fibers may also be important in preventing colorectal cancer. The same properties could be beneficial in subjects who already have IBD. On the other hand, the evidence that dietary fibers can induce relapse is lacking: an RCT [21] of a 2-year high-fiber/low-sugar diet showed no significant benefit or harm in adults with inactive or mildly active CD. Evidence available in UC is conflicting. In the study by Jowett, et al. [22], the rate of relapses was no different between those who avoided fruit and vegetables and dairy products and those who did not. A few studies explored the efficacy of dietary fibers in maintaining remission in UC, reporting an improvement of clinical scores and an additive effect in combination with aminosalicylates [23].

However, vegetables, especially cruciferous ones, contain insoluble fibers that do not absorb water, are metabolically fermented in the colon, and can accelerate bowel movement, triggering symptoms. Another explanation for the role of fibers in exacerbating symptoms could be their content in FODMAPs, through their osmotic effects and fermentation by bacteria. Indeed, unblinded and observational studies have strongly suggested the efficacy of a low FODMAPs diet in controlling functional abdominal symptoms in patients with quiescent IBD [24].

For all these reasons, physicians still advise limiting high-fiber foods, as also reported in the recent paper from the Manchester IBD group on dietary recommendations of healthcare professionals [25]. A lower fiber intake compared to that of the general population in patients with IBD in remission has also been confirmed in two Italian studies [26,27]. These considerations suggest that fibers are perceived as food components that worsen symptoms both by patients and treating physicians. On the basis of existing evidence, a low-fiber diet should be encouraged only in patients with stenosing CD or during flare-ups.

Processed meat was avoided only by 6.6% of our patients, though it has been found that a high consumption of foods containing linoleic acid, red meat, and processed meats increases the risk of developing IBD, especially UC [3,17,18].

One third of our patients adopted a specific diet, most commonly lactose-free, and this was done without performing specific diagnostic texts and without informing the treating physician. In a few cases this type of diet was recommended by the doctor (usually in the flare-up phases based on the hypothesis of lactase deficiency induced by mucosal damage), in other cases it was the patient who made this decision (for example, after consulting specific publications or the web), while in other cases, milk and dairy products were eliminated because they were perceived as foods that caused a worsening of symptoms.

A gluten-free diet was adopted by only 7 patients, 3 of them had a diagnosis of celiac disease. The rate is quite low, since previous studies [10,11] reported a high (27.6%) prevalence of self-reported gluten sensitivity in patients with IBD, especially CD patients with a recent flare-up. A possible explanation could be that most of our cohort were in remission or with mild clinical activity, and gluten sensitivity may be a transient phenomenon for IBD, but age, ethnic origin, educational status, and income also have an impact on the use of gluten-free products.

The use of elimination diets deserves attention because the medical literature has shown that when comparing the data of patients who follow an elimination diet with those who do not, the incidence of malnutrition, even severe, is more frequent in the first group and is associated with a lower intake, especially of calcium, vitamin A, and zinc. Malnutrition translates into a lower response to therapies, while inadequate calcium intake could lead to osteoporosis, particularly in patients on steroids [28,29].

The consequences of the disease and dietary restrictions also have a deep impact on lifestyle and social life: Eating out with family or friends becomes stressful and is often avoided, the pleasure and health benefits of physical exercise are voluntarily given up. Changing diet, trying to find out or understand based on the reactions of the body which foods to avoid and which ones to prefer worsens the sense of frustration that one feels at having to give up foods. Quality of life is further reduced.

This study has some strengths: It was conducted in a relatively large sample of patients with IBD, with homogeneous ethnic origins and food culture; patients were consecutively recruited and are representative of our IBD series. However, the study also has some limitations. The most important could be recall bias, which occurs in most studies investigating the role of nutrition. The questionnaire concerned dietary patterns before and after the diagnosis. The answers to questions formulated by doctors or dietitian could have been influenced by duration of disease and disease activity at the time of interview. Patients were recruited during follow-up visits or scheduled infusion of biologics and were in remission, but disease course before the interview impacts on the perceptions of the role of different factors. The cause–effect relationship of food triggers is difficult to identify and can change with time. Responses and beliefs can also be influenced by co-existing irritable bowel syndrome, a quite common association in IBD in remission [30]. Indeed, in our cohort there was a discrepancy between disease activity as measured by clinical scores and patients’ perception, suggesting the presence of functional symptoms not related to inflammation, which could be linked to food triggers.

Gastroenterologists in the past have paid little attention to the role of diet in the pathogenesis and the course of IBD, but clues from epidemiological studies, suggesting a relevant role of a high fat/high sugar diet and food additives like emulsifiers [3], results from basic research and from interventional trials on specific food components, and newer elimination diets provide a growing body of evidence that should lead to a paradigm change. The contradictory dynamic between health professionals’ advice regarding the lack of a documented effect of diet in IBD and patients’ everyday experiences of eating and drinking must be overcome. Busy physicians are not prone to discuss dietary items during office-visits, but eating is a relevant part of one’s daily life and social interactions, so their attitude for dietary counseling should change since it encourages self-experience. Unfortunately, a high level of uncertainty in dietary advice among health care professionals has recently been confirmed by a survey of the Manchester IBD group: though 48% believed that diet was involved in IBD development and 53% believed that diet could trigger disease relapse, dietary recommendations were highly variable. However, most gastroenterologists were likely to refer patients to a dietitian. Dietitians felt most comfortable in the management of functional gastrointestinal symptoms in quiescent IBD [24].

Further high quality studies with rigorous methodology are warranted to assess the role of food triggers and dietary strategies [31], which should integrate pharmacological therapies. Nutritional and dietary knowledge should be delivered in educational and training programs in IBD management. Dieticians should be always part of the IBD multidisciplinary team to help patients in food choices in order to avoid unnecessary restrictions that cause frustration, impact on quality of life, and might cause malnutrition and specific nutrients deficiencies.

## Figures and Tables

**Table 1 nutrients-13-00759-t001:** Baseline characteristics of the whole study population.

Age (mean ± SD)	48.6 ± 16
Gender	
Male	96 (57.5%)
Female	71 (42.5%)
Marital status	
Single	57 (34.1%)
Married	110 (65.9%)
BMI (mean ± SD)	24.4 ± 3.9
Diagnosis	
UC	81 (48.5)
CD	86 (51.5%)
Smoking habit	
No	70 (41.9%)
Yes	34 (20.4%)
Ex-smokers	63 (37.7%)
Disease duration	
<5 years	56 (33.5%)
5–10 years	44 (26.3%)
11–20 years	34 (20.4%)
>20 years	33 (19.8%)
Disease activity ( patients’ perception)	
remission	66 (39.5%)
mild	45 (26.9%)
moderate	50 (29.9%)
severe	6 (3.6%)
Disease Activity UC (Partial Mayo Score)	
remission	50 (61.7%)
mild	27 (33.3%)
□ moderate	3 (3.7%)
□ severe	1 (1.2%)
Disease activity CD (HBI)	
□ remission	63 (73.3%)
□ mild	12 (14%)
□ moderate	10 (11.6%)
□ severe	1 (1.2%)
Therapy	
Mesalamine	140 (83.8%)
Immunomodulators	24 (14.4%)
Steroids	34 (20.4%)
Biologics	84 (50.3%)
Infliximab	44 (52.4%)
Ustekinumab	2 (2.4%)
Vedolizumab	38 (45.2%)

standard deviation (SD). ulcerative colitis (UC). Crohn’s disease (CD). body mass index (BMI).

**Table 2 nutrients-13-00759-t002:** Comparison between the two groups of patients: patients treated with conventional therapy and patients treated with biologics.

Variable	Conventional Therapy *N* = 83	Biologic Therapy *N* = 84	*p*
Age (mean ± SD.)	51.5 ± 15.6	45.7±16.0	0.02
Sex			
Male	45 (54.2%)	51 (60.7%)	0.4
Female	38 (45.8%)	33 (39.3%)	
Marital status			
Single	25 (30.1%)	24 (28.6%)	0.9
Married/widower/divorced	58 (69.9%)	60 (71.4%)	
Smoker			
No	34 (41.0%)	36 (42.9%)	0.9
Yes (including ex smokers)	49 (59.0%)	48 (57.1%)	
BMI (mean ± SD)	23.5 ± 3.9	25.4±3.7	0.002
Diagnosis			
UC	43 (51.8%)	38 (45.2%)	0.4
CD	40 (48.2%)	46 (54.8%)	
Diet as cause of disease			
No	45 (54.2%)	54 (64.3%)	0.2
Yes	38 (45.8%)	30 (35.7%)	
Dietary changes after diagnosis			
No	15 (18.1%)	15 (17.9%)	0.9
Yes	68 (81.9%)	69 (82.1%)	
Symptoms changes after diet changes			
No	16 (23.5%)	32 (46.4%)	0.005
Yes	52 (76.5%)	37 (53.6%)	
Specific diet			
No	59 (71.1%)	62 (73.8%)	0.7
Yes	24 (28.9%)	22 (26.2%)	

standard deviation (SD).

**Table 3 nutrients-13-00759-t003:** Comparison between patients who perceived and patients who did not perceive diet as the cause of their disease.

Diet Perceived as Cause of the Disease	No (*N* = 99)	Yes (*N* = 68)	*p*
Age (mean ± SD)	48.1 ± 14.5	49.3 ± 18.0	0.6
Gender			
Male	56 (56.6%)	40 (58.8%)	0.8
Female	43 (43.4%)	33 (41.2%)	
Marital status			
Single	25 (25.3%)	24 (28.6%)	0.1
Married/widower/divorced	74 (74.7%)	44 (71.4%)	
Smoker			
No	42 (42.4%)	28 (41.2%)	0.9
Yes (ex smokers included)	57 (57.6%)	48 (58.8%)	
BMI (mean ± SD)	24.8 ± 3.9	23.9±3.9	0.2
Diagnosis			
UC	53 (53.5%)	28 (41.2%)	0.1
CD	46 (46.5%)	40 (58.8%)	
Dietary changes after diagnosis			
No	26 (26.3%)	4 (5.93%)	0.001
Yes	73 (73.7%)	64 (94.1%)	
Changes in symptoms after dietary changes			
No			
Yes	35 (47.9%)	13 (20.3%)	0.001
Follow a specific diet*	38 (52.1%)	51 (79.7%)	
No			
Yes	71 (71.7%)	0 (73.5%)	0.8
Therapy	28 (28.3%)	18 (26.5%)	
Biologics			
Conventional therapy	54 (54.5%)	30 (44.1%)	0.2
	45 (45.5%)	38 (55.9%)	

* lactose or gluten-free or low-residue diet. standard deviation (SD). ulcerative colitis (UC). Crohn’s disease (CD). body mass index (BMI).

**Table 4 nutrients-13-00759-t004:** Comparison between patients who modified their diet and those who did not.

Diet Changed after Diagnosis	No (*N* = 30)	Yes (*N* = 137)	*p*
Age (mean ± SD)	48.0 ± 16.1	48.7 ± 16.0	0.8
Gender			
Male	21 (70.0%)	75 (54.7%)	0.1
Female	9 (30.0%)	62 (45.3%)	
Marital status			
Single	9 (30.0%)	40 (29.2%)	0.9
Married/widower/divorced	21 (70.0%)	97 (71.8%)	
Smoker			
No	14 (46.7%)	56 (40.9%)	0.6
Yes (ex smokers included)	16 (53.3%)	81 (59.9%)	
BMI (mean ± SD)	24.1 ± 3.2	24.5 ± 4.1	0.6
Diagnosis			
UC	17 (56.7%)	64 (46.7%)	0.3
CD	13 (43.3%)	73 (53.3%)	
Diet perceived as cause of the disease			
No	26 (86.7%)	73 (53.3%)	0.001
Yes	4 (13.3%)	64 (46.7%)	
Changes in symptoms after dietary changes			
No		48 (35.0%)	
Yes		89 (65.0%)	
Follow a specific diet			
No	28 (93.3%)	93 (67.9%)	0.005
Yes	2 (6.7%)	44 (32.1%)	
Therapy			
Biologics	15 (50.0%)	69 (50.4%)	0.7
Conventional therapy	15 (50.0%)	68 (49.6%)	

standard deviation (SD). ulcerative colitis (UC). Crohn’s disease (CD). body mass index (BMI).

**Table 5 nutrients-13-00759-t005:** Comparison between patients who adopted specific diets and those who did not.

Specific Diet	No (*N* = 121)	Yes (*N* = 46)	*p*
Age (mean ± SD)	48.2 ± 16.5	49.7 ± 14.8	0.5
Gender			
Male	77 (63.6%)	19 (41.3%)	0.009
Female	44 (36.4%)	27 (58.7%)	
Marital status			
Single	39 (32.2%)	10 (21.7%)	0.2
Married/widower/divorced	82 (67.8%)	36 (78.3%)	
Smoker			
No	56 (46.3%)	14 (30.4%)	0.1
Yes (ex smokers included)	65 (53.8%)	−69.60%	
BMI (mean± SD)	24.5 ± 4.0	24.2 ± 3.9	0.7
Diagnosis			
UC	60 (49.6%)	21 (45.7%)	0.6
CD	61 (50.4%)	25 (54.3%)	
Diet perceived as cause of the disease			
No	71 (58.7%)	28 (60.9%)	0.8
Yes	50 (41.3%)	18 (39.1%)	
Dietary changes after the diagnosis			
No	28 (23.1%)	2 (4.3%)	0.005
Yes	93 (76.9%)	44 (95.7%)	
Changes in symptoms after dietary changes			
No	35 (37.6%)	13 (29.5%)	0.3
Yes	58 (62.4%)	31 (70.5%)	
Therapy			
Biologics	62 (51.2%)	22 (47.8%)	0.7
Conventional therapy	59 (48.8%)	24 (52.2%)	

standard deviation (SD). ulcerative colitis (UC). Crohn’s disease (CD).

**Table 6 nutrients-13-00759-t006:** Foods perceived as a symptoms’ trigger by patients with inflammatory bowel disease (IBD).

Frequently Avoided Foods	*N* = 167
Spicy foods	82 (49.1%)
Seasoned foods	64 (38.3%)
Fried foods	48 (28.7%)
Carbonated drinks	50 (29.9%)
Milk and dairy products	57 (34.1%)
Energy drinks	12 (7.2%)
Alcoholic drinks	31 (18.6%)
Pork meat	11 (6.6%)
Processed meat	11 (6.6%)
Vegetables	47 (28.1%)
Fruit	27 (16.2%)
Legumes	32 (19.2%)
Whole grain bread	22 (13.2%)
Bread	8 (4.8%)
Eggs	4 (2.4%)
Rice	3 (1.8%)
Chicken	1 (0.6%)
Pasta	2 (1.2%)
Fish	1 (0.6%)
Coffee	21 (12.6%)
Refined sugars (sweets)	15 (9%)

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
