# Peer review of "Perception of the Role of Food and Dietary Modifications in Patients with Inflammatory Bowel Disease: Impact on Lifestyle"

_nutrients, 2021, doi:10.3390/nu13030759_

Round 1

Reviewer 1 Report

This study addresses an important issue as patients almost always express an interest in the role of diet in initiating, controlling or treating their IBD and also in modifying it after the diagnosis or when inn a disease flare.

There are several areas that need to be addressed:

  1. Numerous other studies have been conducted and particularly recently by the Manchester IBD group (JK Limdi) that have not been referenced in the introduction. These include:
    1. Crooks B et al. Eur J Gastroenterol Hepatol. 2021 Mar 1;33(3):372-379. doi: 10.1097/MEG.0000000000001911. PMID: 32956176.
    2. Limdi JK et al. Inflamm Bowel Dis. 2016 Jan;22(1):164-70. doi: 10.1097/MIB.0000000000000585. PMID: 26383912.
  1. How was your questionnaire developed? Did it go through a patient focus group?
  2. Your discussion needs to reference your findings and contextualise the similarities and/or differences with previous work, some of which you have referenced and many other studies such as the papers quoted above which have not been included. including Jowet et al, Cohen et al etc. 
  3. How do you relate the findings to then presence of concomitant IBS?
  4. You refer to advice given by health care professionals and discuss the role of dieticians. Again, the Manchester IBD group has recently published a study from a UK-wide group of health care professional perspective of diet and advice given.
    1. Ref:Crooks B, McLaughlin J, LimdiDietary beliefs and recommendations in inflammatory bowel disease: a national survey of healthcare professionals in the UK. 
    2. This should be included and suitably referenced along with other work in this area.

Thank you for considering these comments and revising your manuscript accordingly.

A recent review article from the same group may be useful for additional references:Limdi JK. Dietary practices and inflammatory bowel disease. Indian J Gastroenterol. 2018 Jul;37(4):284-292. doi: 10.1007/s12664-018-0890-5. Epub 2018 Sep 12. PMID: 30209778; PMCID: PMC6153885.

Reviewer 2 Report

This is a study out of Italy that looks at the correlation between IBD diagnosis and its impact on food consumption/aversion. Given the high degree of association between food and gastrointestinal symptoms, an interesting look at patients' food practices is quite interesting. The study itself is sound, questionnaires shown, however am not able to see the results tables 1-4 in figure or supplemental. It is unclear if this is typesetting error from MDPI or no uploading from authors. The changes food practices are intriguing especially the gluten-free diet finding.

Some suggestions:
1) Recheck tables/graphs are uploaded, again, am not sure if this is an error of the journal uploading

2) Moderate language revision would help, particularly in the discussion "things are changing" should not be written in a discussion format. Would recommend some language changes, particularly to discussion

3) Graph of the practice changes, or some sort of pie charts displaying the practice changes in diet would help.

Look forward to next revision.

Reviewer 3 Report

This is an interesting piece of research in a population that the authors say is under-represented in previous research on this topic. The authors make a strong argument for clinicians to consider patient views on diet and nutrition and to also consider how dietary changes implemented by patients may affect their health holistically. 

General comments. There are large chunks of text that need to be referenced. These include the points made in the following lines:

Line 36 - 37

Lines 43 - 50

Lines 52-54

Lines 207-213

Lines 215 - 220

The introduction was relevant and set the scene but needs to adequately reference the existing literature as suggested above.

The methods section was adequate but could be improved be considering the following. 

The first sentence includes data that is part of the results. I recommend only describing the methods - e.g. Patients were recruited from ..... clincis from September 2019 etc.

The sentence "The interview lasted 20 - 30 minutes" would fit better higher up when the questionnaire is described. It is not obvious that it is an interview based data collection until the end last sentence.

Line 89-90 is a repeat of the first sentence of the methods and is a result rather than a method. 

Line 92 - who completed the Partial Mayo Score and HBI? A physician, self reported or the interviewer.

Line 94 - suggest putting BMI after weight and height given that a BMI cannot be calculated without the weight and height first.

Line 96 - what about the rest of the data, how was that analysed e.g. as a whole group or in the two separate groups?

Line 97 - which patients are in the two groups? Is one group those from the outpatient clinic and the other from the infusion clinic? It is not clear. 

Line 98 - please add further information about the version of SPSS and the software company etc. 

Results section is fine but it was difficult to review it fully because I was not provided with Table 1. In general it appears that much of the information that would be in Table 1 was written in full in the text. I recommend just highlighting the main findings in the text. 

Line 104 - how many patients were approached but did not consent to take part? Please add this information.

Line 121 - I was unsure of who was in the two groups. This may have been more clear from Table 1 but also if it was adequately described in the methods section.

Discussion tells a cohesive story but lacks references to the literature to support the authors statements (as indicated above). My specific comments on the discussion are:  

Line 168 - I recommend using "proportion" rather than "rate"

Line 186 - "abolition of fibers" I don't think that abolition is the right word in english and recommend revising this wording.

Line 186 - 188 - I disagree with this line of thinking. Is it relevant that epidemiological evidence suggests that a low fibre increases their risk of developing IBD if the patient cohort already has IBD? I agree that a low fibre diet is not indicated in patients with quiescent IBD but the reason for this is not because it increases their risk of developing the disease they already have. This paragraph would fit better if the authors discussed the evidence for fibre and the by-products of fibre digestion by gut microbes on potentially reducing risk of disease relapse. This paragraph also needs references added.

Line 213 - is it calcium deficiency or inadequate calcium intake that is associated with increased risk of osteoporosis? 

Round 2

Reviewer 1 Report

Thank you for addressing reviewer comments and improving your manuscript accordingly.

Reviewer 2 Report

Authors have addressed my concerns. Tables and charts are also present.